# Assessing household financial burdens for preprimary education and associated socioeconomic inequalities: a case study in China

Yi Wei,[1] Kai Liu,[2] Le Kang,[3] Jere R Behrman,[4] Linda M Richter,[5] Alan Stein,[6] Yingquan Song,[1] Chunling Lu  [7,8]

For numbered affiliations see end of article.

**Correspondence to**
Dr Yingquan Song;
songyingquan@pku.edu.cn

## ABSTRACT

**Background** Providing young children with universal access to preprimary education (PPE) is considered a powerful tool for human capital development and eliminating the intergenerational transmission of poverty. To remove household financial barrier for achieving universal PPE, this study proposed a measure to identify households incurring 'heavy financial burdens from paying for PPE' (HBPPE) and conducted a case study in China.

**Methods** Using nationally representative data in 2019, we estimated the percentage of households with HBPPE (spent 7% or more of their total annual expenditure) and associated socioeconomic inequalities. We also applied a three-level logit regression model to investigate the factors associated with the probabilities of households incurring HBPPE.

**Results** Half of the sampled households spent 7% or more of their expenditures on PPE. Households in the lowest wealth quintile (54%) or households with children attending private PPE (55%) had higher percentages of HBPPE than households in other wealth quintiles (eg, 51% in the highest wealth quintile) or households with children attending public kindergartens (41%). Logit regression analysis shows that the poorest households and households with children attending private kindergarten were more likely to incur HBPPE than their counterparts.

**Conclusion** To ensuring universal access to PPE in China, future policy should consider increasing the enrolment of children from low-income families in public kindergartens and increasing governmental investments in low-income households by subsidising children attending PPE.

## INTRODUCTION

Preprimary education (PPE) has important influences on the development of cognitive and non-cognitive skills of children, especially for those from disadvantaged groups.[1] The high return on investment in PPE makes it an important strategy for strengthening national human capital and reducing inequality and the intergenerational transmission of poverty.[2–8] Quality PPE for all girls and boys has been included in the 2030 Sustainable Development Goals.[9]

### WHAT IS ALREADY KNOWN ON THIS TOPIC

⇒ Significant socioeconomic inequalities in preprimary education (PPE) attendance were observed in many low-income and middle-income countries (LMICs).
⇒ Empirical evidence on household financial barrier to PPE is very limited in LMICs. In China, disadvantaged households (eg, low income, low maternal schooling) paid larger share than their counterparts for sending children to preschool. Little, however, is known about how many households suffered from heavy financial burden from paying for PPE.

### WHAT THIS STUDY ADDS

⇒ This study proposed a measure to identify households incurring 'heavy financial burdens from paying for PPE' (HBPPE) and conducted a case study in China using nationally representative data.
⇒ Findings showed that almost 50% of the sampled households incurred HBPPE (spent over 7% or more of their annual expenditures on PPE), with the poorest households and households in which children attended private kindergarten more likely to incur HBPPE than their counterparts.

### HOW THIS STUDY MIGHT AFFECT RESEARCH, PRACTICE OR POLICY

⇒ This study proposed a measure on HBPPE that could be used for similar research in understanding a country's progress in eliminating the financial barrier to universal coverage of PPE.
⇒ Findings of this study could inform policy-makers and other stakeholders in China on who suffered HBPPE (eg, the poorest households or households with children attending private kindergarten) and what could be potentially effective policy instruments in promoting equal access to PPE.

Ensuring universal access to quality PPE requires sufficient investment and effective design of cost-sharing mechanisms between governments and households. In many low-income and middle-income countries (LMICs), government's financial support to PPE remains rather limited,[10–12] making

it difficult for the poor households accessing PPE. In Nigeria, for example, less than 10% of the poorest young children attended PPE in 2010, compared with approximately 80% of the richest young Nigerian children.[13]

China has the second-largest child population in the world and has set a policy goal of universal coverage of PPE in 2010.[14] Since then, public expenditures on PPE have increased rapidly from ¥24.44 billion (1.67% of governmental educational spending) in 2010 to ¥200.8 billion (5.01% of governmental educational spending) in 2019 (online supplemental figure A1).[15] The gross enrolment rate for 3-year PPE also grew rapidly from 46.1% in 2000 to 83.4% in 2019 (online supplemental figure A2).[15]

PPE in China is not free. Households have to pay tuition fees that vary greatly by type of kindergarten, public or private. Public kindergartens are essentially 'club goods', which are the extension of the welfare system in the era of planned economy, as a benefit provided by organisations to their 'employees'. These public kindergartens are not open to the public and have certain requirements for admission, for example, children of employees of governments, universities or state-owned enterprises, etc children from more-advantaged groups are more likely to enter this kind of public kindergarten with relatively low fees and higher quality.[16 17] Currently, government spending on PPE mainly goes to these public kindergartens, especially the kindergartens run by educational departments and other governmental agencies.

Private kindergartens have become an important part of China's PPE system in recent years, with 56% of young children enrolled in private kindergarten in 2019.[15] The majority of private kindergartens have lower quality and are more expensive than public kindergartens. They are used by children who do not have access to public ones. The major share (85%) of the funding in private kindergartens came from tuition fees contributed by households, compared with 25% in public kindergartens.[18]

Limited empirical evidence has shown that disadvantaged households usually paid larger share of their income on PPE than their counterparts in China.[12 19 20] However, little is known regarding how many households suffered from heavy financial burdens from paying for PPE (HBPPE). To fill in this knowledge gap and help policy-makers identify those who need support most, this study defined HBPPE using the approach adopted by the World Bank (WB) and the WHO in defining catastrophic household health spending.[21 22] For example, the WB defines a household having catastrophic health spending if it spends 10% or more of its total annual expenditures on health. The 10% cut-off has been widely used as an important policy measure for tracking progress towards universal health coverage. This study proposes a similar threshold for HBPPE so as to better understand household financial burdens for paying for PPE and hold governments accountable for their promise of 'quality PPE for all girls and boys'. If 10% of a household's spending on health has been accepted as 'catastrophic' to the household, the same definition could be applied to PPE payments too. We therefore proposed a measure to identify households incurring HBPPE and conducted a case study in China. Using the 2019 China Household Finance Survey (CHFS), we assessed the national-level and provincial-level prevalence of HBPPE and associated socioeconomic and inequalities.

## METHODS
We used the Consolidated Health Economic Evaluation Reporting Standards reporting guidelines in the writing of this paper.[23]

### Defining households with heavy financial burdens for paying for PPE (HBPPE)
There is no available measure for catastrophic household PPE spending. Following the practice of the WB and the WHO in defining catastrophic household health spending, we defined a household incurring HBPPE if it spends 7% (10% or 20%) or more of its total annual expenditures on PPE. The selection of these thresholds was based on the definition of catastrophic health spending as well as a current federal benchmark for households receiving subsidies on PPE proposed by the Office of Child Care under the US Department of Health and Human Services, stating that households receive subsidies if their out-of-pocket spending on PPE exceed 7% (previously 10%) of households' incomes.[24] Though the threshold of 7% comes from the US policy on subsidising PPE, we think that it could also be applied to the context of China as the cut-off is in terms of proportion of income rather than actual income. If households who spent 7% or more on PPE need to be subsidised in high-income countries, it should also be the case in LMICs considering that households in LMICs have less capacities to pay compared with those in high-income countries.

### Data and samples
We used nationally and provincially representative data from the 2019 CHFS that collects information on household assets, income, wealth, consumption, as well as socioeconomic and demographic characteristics of the sampled households and their members.[25] The CHFS employs a stratified three-stage random sample design (details in online supplemental chapter 1 and figure A1). The 2019 CHFS data is publicly accessible and covers 29 provinces (excluding Tibet, Xinjiang, Hong Kong and Macau), 345 cities/districts/counties, 34643 households and 107008 individuals. Our study focused on households with children attending kindergarten, including those under age 3 or over age 6 who still attended kindergarten (18.2% of the sample), because the main purpose of this study is to identify households incurring HBPPE. The final sample includes 3371 young children attending kindergartens, from 3111 households (9% of the total sample), who reported available PPE expenditure data.

## Measuring households with HBPPE

To assess if household suffered from HBPPE, we calculated estimates of household spending on PPE and total expenditures in the previous 12 months. Caregivers reported household expenditure on kindergarten over the 12 months prior to the survey. The expenditure can be categorised into two groups: (1) fees charged by schools (in-school expenditures) including tuition fees, books and materials, uniforms, meals, transport and accommodation and (2) fees on out-of-school education (out-of-school expenditure) including after-school tutoring (academically focused), extracurricular activities (eg, learning piano or sports) and home-learning materials (eg, books, computers). To estimate the sensitivity of the findings to the definition of PPE spending,

we excluded out-of-school spending from PPE spending in an alternative analysis.

The CHFS includes a series of questions on household consumption over the 12 months prior to the survey that cover spending on food, clothing, housing, daily necessities and service, healthcare, transportation and communication, education and entertainment, and others (eg, legal service). We obtained the annual household expenditure by summing spending on these items. For each household, we calculated the percentage of household total expenditure devoted to PPE. Based on the cut-off percentage value (7%, 10% or 20%), we then constructed a dichotomous variable indicating a household's status of HBPPE: if a household spent more than 7% (10% or

| Table 1 | Summary statistics of variables used in the analysis | | | | | |
|---|---|---|---|---|---|
| Variable | Observation | Mean | SD | Min | Max |
| Child-level variables | | | | | |
| PPE expenditure per child | 3371 | 8220 | 10088 | 0.000 | 204000 |
| Female | 3371 | 0.470 | 0.499 | 0.000 | 1.000 |
| Private kindergarten | 3371 | 0.560 | 0.496 | 0.000 | 1.000 |
| Household-level variables | | | | | |
| PPE expenditure per household (¥) | 3111 | 8907 | 10824 | 0.000 | 204000 |
| Per cent of household expenditure on PPE | 3111 | 0.094 | 0.096 | 0.000 | 0.887 |
| HBPPE (7%) | 3111 | 0.495 | 0.500 | 0.000 | 1.000 |
| HBPPE (10%) | 3111 | 0.342 | 0.475 | 0.000 | 1.000 |
| HBPPE (20%) | 3111 | 0.098 | 0.297 | 0.000 | 1.000 |
| Number of children undertaking PPE | 3111 | 1.084 | 0.294 | 1.000 | 4.000 |
| Children in private kindergarten | 3091 | 0.557 | 0.497 | 0.000 | 1.000 |
| Household size | 3111 | 5.131 | 1.576 | 2.000 | 15.000 |
| Per capita household net wealth (¥) | 3111 | 241004 | 562588 | −1974739 | 17020206 |
| Household net wealth quintile | 3111 | | | | |
| Quintile 1 | 3111 | 0.210 | 0.407 | 0.000 | 1.000 |
| Quintile 2 | 3111 | 0.252 | 0.434 | 0.000 | 1.000 |
| Quintile 3 | 3111 | 0.232 | 0.422 | 0.000 | 1.000 |
| Quintile 4 | 3111 | 0.187 | 0.390 | 0.000 | 1.000 |
| Quintile 5 | 3111 | 0.119 | 0.324 | 0.000 | 1.000 |
| Maternal schooling | | | | | |
| Primary school and below | 3111 | 0.136 | 0.343 | 0.000 | 1.000 |
| Middle school | 3111 | 0.436 | 0.496 | 0.000 | 1.000 |
| High school | 3111 | 0.180 | 0.384 | 0.000 | 1.000 |
| College or above | 3111 | 0.248 | 0.432 | 0.000 | 1.000 |
| Rural residential area | 3111 | 0.352 | 0.478 | 0.000 | 1.000 |

The number of children undertaking PPE indicates the number of children who attended kindergarten in a household. Household size is defined as the total number of household members in a household. A household unit is defined as all members of a housing unit related by blood, marriage or some other legal arrangement; or two or more persons use incomes to make joint expenditures; or a single person living with others but is financially independent. Thus, a household may include grandparents, especially in a rural area. When analysing households that have children in private kindergarten, we excluded the observations of 20 households that had children attending both public and private kindergarten

HBPPE, heavy financial burdens from paying for PPE; PPE, preprimary education.

20%) of its total expenditure on PPE, we classified the household as experiencing HBPPE.

### Measuring socioeconomic disparities in HBPPE

We assessed the disparities in the percentage of households with HBPPE by household economic status (five wealth quintiles), maternal schooling (primary school or below, middle school, high school, and college or above), residential area (rural vs urban) and type of kindergarten (public vs private). Disparity in this study was quantified as the absolute differences between the two subgroups. Details on constructing these variables are presented in online supplemental chapter 2.

### Statistical analysis

We first performed a descriptive analysis by assessing the level and socioeconomic inequalities in PPE spending per child and percentages of households with HBPPE. All estimates were adjusted for sampling weights (individual and household weights), primary sampling unit (county) and stratification variable (province). We estimated the statistical significance of inequalities in PPE spending and % of households with HBPPE between the two subgroups (eg, rural vs urban) (details in online supplemental chapter 3).[26]

Considering the large variation in economic development and geographical situations across the provinces, we assessed the provincial-level disparities in the level of PPE spending per child and % of households with HBPPE across the 29 provinces, using the gap between the weighted mean of each province from the national mean (details are provided in online supplemental chapter 4).

To identify the factors associated with the probabilities of a household incurring HBPPE, we used a three-level logit regression model with random intercepts by province (level 3) and county (level 2), controlling for potential clustering effects at the province and county levels (Eq 1).

$$Logit\ (Y_{ijk}) = \beta_0 + \beta X_{ijk} + u_{0k} + v_{0jk} + \varepsilon_{0ijk} \qquad (1)$$

where $Logit(Y_{ijk})$ represents the probability of incurring HBPPE in the $i^{th}$ household who lives in county j and province k. $\beta$ is a vector of coefficients for $X_{ijk}$ which is a vector of variables with individual or household characteristics, including the number of children undertaking PPE, children attending private kindergarten, household size, maternal schooling, wealth quintile and residential area. $u_{0k}$ and $v_{0jk}$ represent between-province random variations and between-county/within-province random variations, respectively. $\varepsilon_{0ijk}$ represent random errors. Summary statistics for these variables are presented in table 1.

Regression analyses were conducted using three different thresholds for HBPPE: 7%, 10% and 20%. To further test the sensitivity of the findings, we also used the proportion of household consumption on PPE as an outcome variable and conducted descriptive and regression analysis, as shown in online supplemental chapter

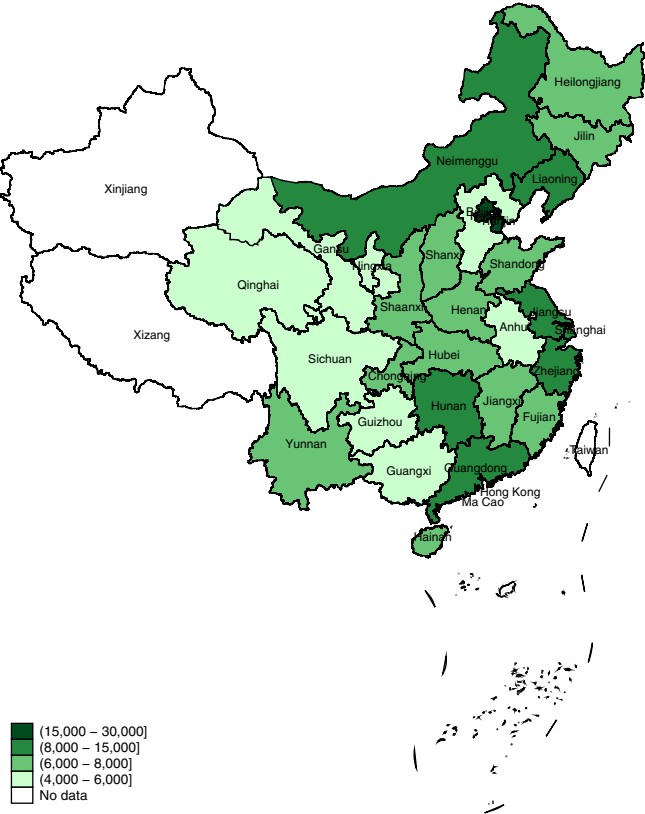

**Figure 1** Average level of PPE spending per child across 29 provinces.

5. In addition, we also used continuous household net wealth instead of net wealth quintiles as a predictor and conducted regression analysis. The STATA programme (V.16.1) was used in the analyses.

## RESULTS

Among all the children attending kindergarten, 47.0% were girls and 56.0% attended private kindergarten. Among the total of 3111 households, 35.2% lived in a rural areas; 11.9% or 21.0% were in the highest or lowest wealth quintile, respectively; 24.8% of the mothers held college degrees or higher, whereas over 57.2% less than a high-school diploma (table 1).

### Level and inequalities in household PPE spending per child

On average, PPE spending per child at the national level was ¥8220 (table 1), ranging from ¥4070 in Qinghai province to ¥23 348 in Beijing (figure 1 and table 2). Well-developed provinces had much higher PPE spending per child than the under-developed central and western provinces (table 2).

For households sending children to private kindergartens, or in urban areas, or with college-educated mothers, or from the highest wealth quintile, their PPE spending per child was significantly higher than that of their counterparts (figure 2). For example, PPE spending per child at the private kindergartens was approximately 30% higher higher than that at public kindergartens (¥8075

**Table 2** Average level of PPE spending per child across 29 provinces and weighted mean difference of the mean between the provincial level and the national level

| Province | PPE spending at the provincial level | Weighted mean difference* |
|---|---|---|
| Beijing | 23348 | 246 |
| Tianjin | 19487 | 135 |
| Liaoning | 13417 | 189 |
| Zhejiang | 11057 | 155 |
| Shanghai | 10279 | 51 |
| Neimenggu | 10095 | 50 |
| Guangdong | 9309 | 162 |
| Jiangsu | 8163 | 47 |
| Hunan | 8030 | 34 |
| Henan | 7942 | 41 |
| Fujian | 7877 | 15 |
| Shaanxi | 7740 | 11 |
| Chongqing | 7062 | −6 |
| Heilongjiang | 6980 | −10 |
| Hubei | 6964 | −16 |
| Shandong | 6852 | −35 |
| Jilin | 6850 | −10 |
| Hainan | 6614 | −5 |
| Jiangxi | 6202 | −38 |
| Shanxi | 6090 | −33 |
| Yunnan | 6004 | −47 |
| Ningxia | 5496 | −9 |
| Sichuan | 5456 | −113 |
| Anhui | 4869 | −113 |
| Guizhou | 4674 | −69 |
| Guangxi | 4536 | −100 |
| Gansu | 4506 | −54 |
| Hebei | 4246 | −168 |
| Qinghai | 4070 | −14 |

*The measures took zero as the reference, with positive values indicating provincial means higher than the national average and negative values indicating provincial means lower than the national average.
PPE, preprimary education.

vs ¥6193). PPE spending per child in the richest households was four times more than that in the poorest households (¥17 007 vs ¥4182) (statistical testing is shown in online supplemental table A1).

When excluding out-of-school expenditure from PPE spending, the main results remain robust (online supplemental figure A4).

## Level and inequalities in the percentage of HBPPE
Approximately 49.5% of households at the national level spent more than 7% of their total annual expenditure on paying for PPE, our initial threshold for HBPPE. This percentage reduced to 34.2% or 9.8%, respectively, if the threshold was set at 10% or 20% (table 1). Provinces in the Northeastern and Middle areas of China had higher percentages of HBPPE than other areas, with Liaoning province (76.9%) the highest (figure 3 and online supplemental table A2). When using alternative thresholds (10% or 20%), the percentage of households with HBPPE in each province was reduced, but the pattern across the provinces remained stable (online supplemental figures A5 and A6 tables A3 and A4).

In terms of socioeconomic inequalities (figure 4), per cent of HBPPE was higher among those households sending children to private kindergartens (55%) than those to public kindergartens (41%). The disparity remained significant when using alternative thresholds (online supplemental figures A7 and A8 and tables A5–A7) or when excluding out-of-school spending from PPE spending (online supplemental figures A9–11). Per cent of HBPPE among the richest households was significantly lower than that among the poorest households when using a threshold of 7% or 10%, after excluding out-of-school spending from PPE spending.

### Factors associated with household HBPPE
Regression results (table 3) have identified the following factors associated with HBPPE, across the three thresholds: households in the higher wealth quintiles or with smaller size were less likely to incur HBPPE than their counterparts; households sending children to private kindergarten or with more children in kindergarten were more likely to incur HBPPE than their counterparts.

The results from the sensitivity analysis using the proportion of household spending on PPE as outcome variables are presented in the online supplemental figures A12–A14, tables A8–A10. We used both a multilevel linear regression and a beta regression for estimation. As online supplemental tables A9 and A10 show, the results are consistent with the main regression results.

The results from the sensitivity analysis using continuous household net wealth as a predictor are presented in the online supplemental tables A11 and A12. We used a multilevel logit regression, multilevel linear regression and a beta regression for estimation. As online supplemental tables A11 and A12 show, the results are consistent with the main regression results.

## DISCUSSION
This study proposed a measure to identify households incurring 'HBPPE' and applied the measure to the data in China. Our findings showed that, in China, approximately 50% (34%, 10%) of the households with young children attending kindergarten suffered from HBPPE, when a threshold of 7% (10%, 20%) of their annual expenditures on PPE was applied. Households from provinces in the Northeastern and Middle regions had higher percentages of HBPPE on average than those from other

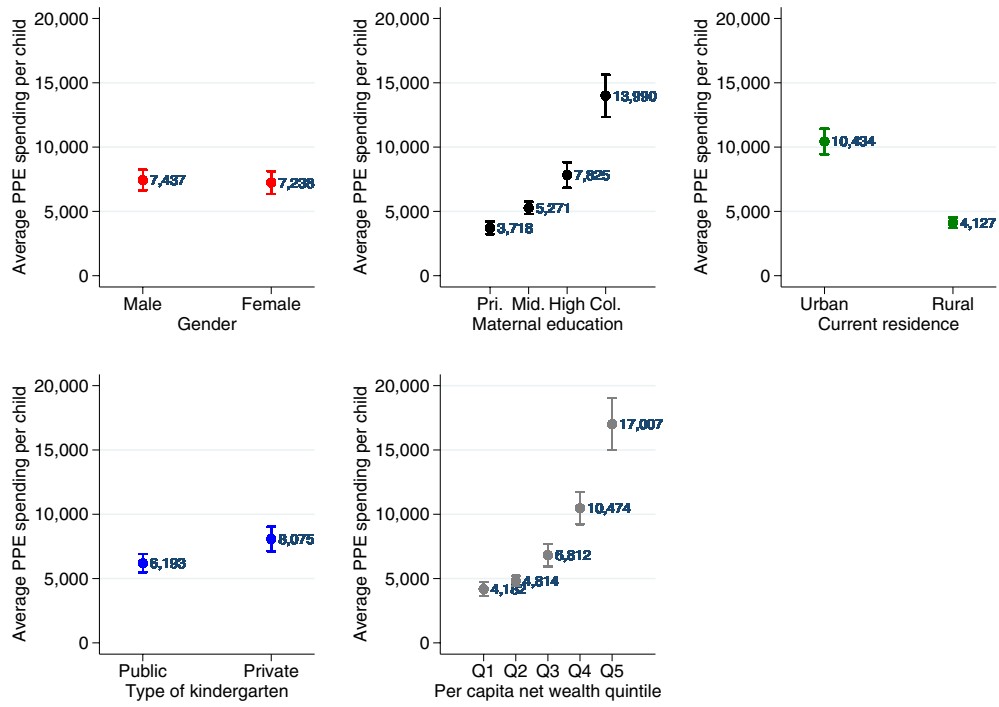

**Figure 2** Level and inequalities of PPE spending per child by gender, type of kindergarten, maternal education, household net wealth quintile, and current residential area.

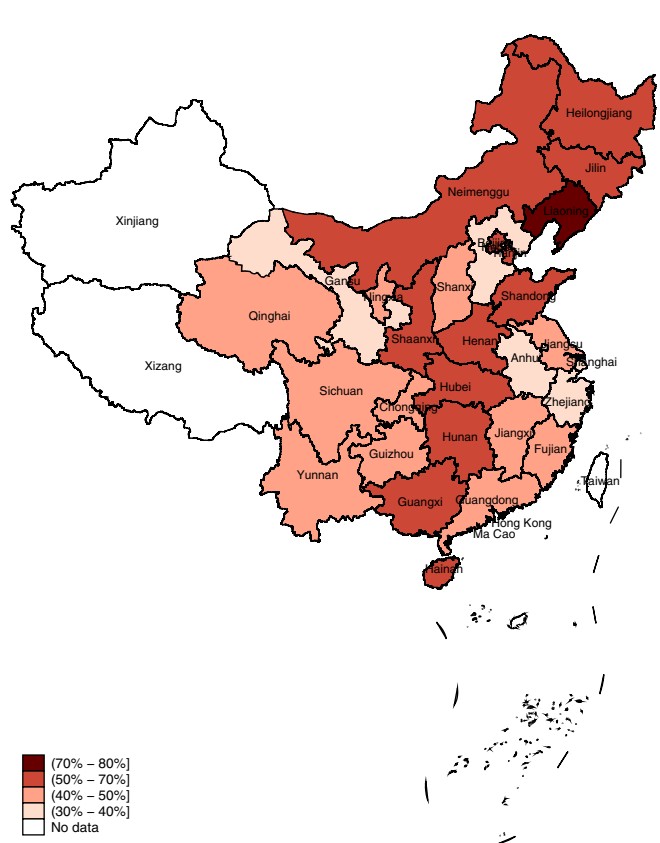

**Figure 3** Level of % of households with HBPPE (threshold=7%) across 29 provinces.

areas. Holding everything else constant, households in the lowest-wealth quintile or sending children to private kindergarten incurred the higher probability of HBPPE than their counterparts.

These findings are consistent with previous studies that indicate variation of PPE spending by household income levels and type of kindergartens.[12 19 20] Our study has further identified significant disparities in household with HBPPE, and found that households with children attending private kindergartens had an increase of 88%–105% (across the three thresholds) in the odds of incurring HBPPE compared with households with children attending public kindergartens. The richest households could have a reduction of 37%–50% in the odds of incurring HBPPE compared with the poorest households. The finding of regional disparities in HBPPE is consistent with empirical evidence regarding the slowdown of economic development in the Northeastern and Middle Regions and their lower percentage of public spending on compulsory education than other regions and on PPE in particular.[27–30]

To ensure universal PPE, the Chinese government has initiated the policy of increasing the supply of public kindergartens and regulating the tuition fees of private kindergartens. As shown in our study, even with the expansion of public kindergartens in recent years, more than half of young children still had to go to expensive private kindergarten. To reduce the financial burden of PPE for households, the Chinese government needs to

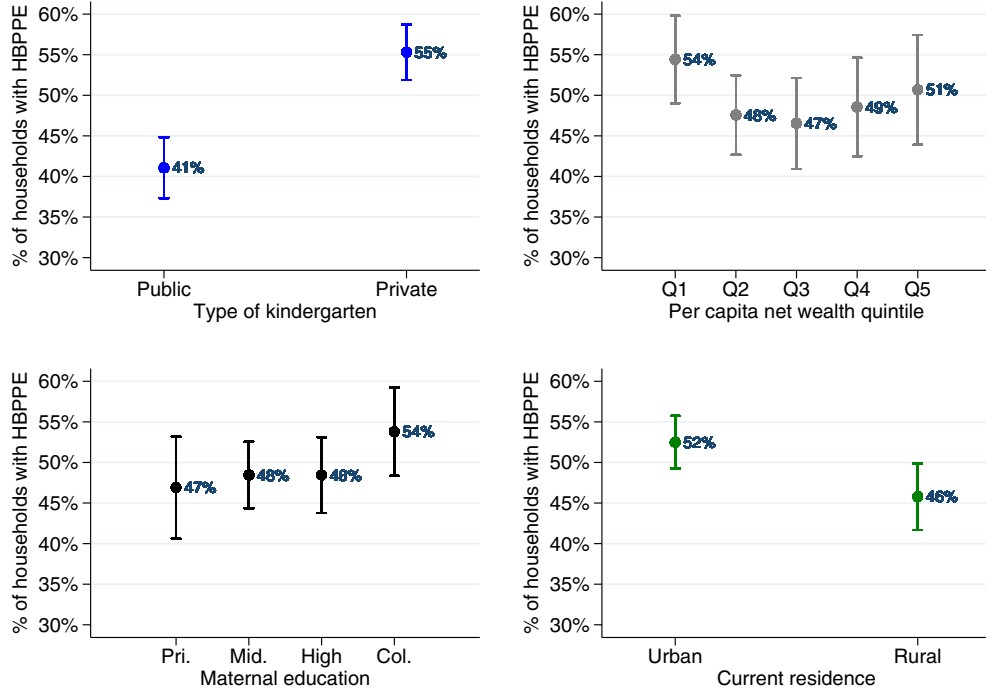

**Figure 4** Level and inequalities of % of households with HBPPE (threshold=7%) by type of kindergarten, maternal education, household net wealth quintile, and current residential area.

continue increasing the public kindergartens and make them accessible to everyone. Meanwhile, providing PPE subsidies to low-income households, especially those with children going to the private kindergartens, is necessary for ensuring the low-income households can affordable PPE.

This study has several limitations. First, the study only included children enrolled in kindergarten at the time of the survey. Children who were not able to attend kindergarten were not included in the analysis sample and they were more likely to live in poor or remote rural areas with limited or even no access to kindergarten. Excluding those children could lead to an underestimation of the probability of HBPPE among the poorest or rural households. Second, the study only used cross-sectional data. With a new wave of survey data available in the future, we will be able to identify the impact of related governmental policies on HBPPE over time. Third, the study used three thresholds to measure households with HBPPE. However, there is no information from households on what percentage of their total spending on PPE would be considered affordable. Fourth, when applying a threshold of HBPPE to households with different income levels, it may underestimate the burden of PPE borne by low-income households because they have less capacity to pay compared with those in the middle or upper quintiles.

Despite the limitations, this study proposed a measure that can be used by researchers to identify households with financial barrier for accessing PPE in LMICs. Using this measure in China, the study provides solid evidence to inform policy-makers on who suffered HBPPE and what could be potentially effective policy instruments in promoting equal access to PPE. Future PPE policy in China should consider establishing a financial-aid system on PPE for the children who need it most and expanding the enrolment of children from disadvantaged households in the public kindergarten system.

**Author affiliations**
[1]China Institute for Educational Finance Research, Peking University, Beijing, China
[2]School of Labor and Human Resources, Renmin University of China, Beijing, China
[3]Institute of Education, Nanjing University, Nanjing, Jiangsu, China
[4]Department of Economics, University of Pennsylvania, Philadelphia, Pennsylvania, USA
[5]DSI-NRF Centre of Excellence in Human Development, University of the Witwatersrand Johannesburg, Johannesburg, Gauteng, South Africa
[6]Department of Psychiatry, Oxford University, Oxford, UK
[7]Division of Global Health Equity, Brigham and Women's Hospital, Boston, Massachusetts, USA
[8]Department of Global Health and Social Medicine, Harvard Medical School, Boston, Massachusetts, USA

**Contributors** YW was responsible for survey coordination, conducted data analysis and wrote the first draft of the manuscript, and was responsible for the overall content as the guarantor. LK drew the map that demonstrates the sample distribution in the survey as well as the financial burdens in preprimary education (PPE) at the provincial level. KL conducted data analysis on inequalities and regressions and took responsibility for the integrity of data analysis. YS was the PI for expenditures of PPE, survey design, initiated the study, supervised YW's work. CL conceptualised the study, designed the analysis plan and mentored data analysis. All authors have made contributions to data interpretation and manuscript writing. All authors approve the manuscript for submission.

**Table 3** Multilevel logit regression results on factors associated with household status on HBPPE defined with various thresholds

|  | HBPPE_7% (OR) | HBPPE_10% (OR) | HBPPE_20% (OR) |
|---|---|---|---|
| Household net wealth quintile (ref. quintile 1) | | | |
| Quintile 2 | 0.833* | 0.756** | 0.665** |
| Quintile 3 | 0.560*** | 0.584*** | 0.581*** |
| Quintile 4 | 0.598*** | 0.697*** | 0.595** |
| Quintile 5 | 0.634*** | 0.512*** | 0.503*** |
| Mother's education (ref. primary school and below) | | | |
| Middle school | 1.192 | 1.200 | 1.095 |
| High school | 1.004 | 1.127 | 1.146 |
| College/above | 1.208 | 1.391* | 1.428 |
| Number of children undertaking PPE | 3.451*** | 3.375*** | 3.233*** |
| Household having children in private kindergarten | 1.973*** | 2.050*** | 1.880*** |
| Household size | 0.799*** | 0.766*** | 0.789*** |
| Rural | 0.820** | 0.892 | 1.225 |
| Constant | 0.744 | 0.407*** | 0.061*** |
| N | 3091 | 3091 | 3091 |
| ICC (province) | 0.018 | 0.029 | 0.048 |
| ICC (county \| province) | 0.110 | 0.108 | 0.187 |

ICC (province) denotes the ICC at the province level, and ICC (county | province) denotes the ICC at the county and province level. When we define the HBPPE measure with the threshold of 7%, for example, we estimated that province random effects composed 1.8% of the total residual variance of the HBPPE measure, while county and province random effects composed 11.0% of the total residual variance of the HBPPE measure.
*p<0.10, **p<0.05, ***p<0.01.
HBPPE, heavy financial burdens from paying for preprimary education; ICC, intraclass correlations; PPE, preprimary education.

**Funding** This work was supported by the Ministry of Education in China (BFA210073), the Ministry of Education in China (7122900028), UNICEF China Office (8430500872) and the UKRI GCRF Harnessing the Power of Global Data to Advance Young Children's Learning and Development (ES/T003936/1).

**Competing interests** None declared.

**Patient and public involvement** Patients and/or the public were not involved in the design, or conduct, or reporting, or dissemination plans of this research.

**Patient consent for publication** Not applicable.

**Ethics approval** We used a secondary data that has been published and publicly accessible.[25] Identifiers for individuals and families have been removed for public use. Users cannot identify individuals and households from this survey dataset. This study is an observational study based on a public dataset instead of a clinical study. Therefore, a patient and public involvement (PPI) statement is not applicable to this study.

**Provenance and peer review** Not commissioned; externally peer reviewed.

**Data availability statement** Data are available in a public, open access repository. Data from the China Household Finance Survey is available at the website of the Survey and Research Center for China Household Finance, at https://chfs.swufe.edu.cn/. The analysis code is available from the corresponding author upon request.

**ORCID iD**
Chunling Lu http://orcid.org/0000-0002-4780-9451

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
