## [Reviewer comments · BMJ Paediatrics Open]

ARTICLE DETAILS

TITLE (PROVISIONAL)	Assessing household financial burdens for preprimary education and associated socioeconomic inequalities – a case study in China
AUTHORS	Wei, Yi Liu, Kai Kang, Le Behrman, Jere Richter, Linda Stein, Alan Song, Yingquan Lu, Chunling

VERSION 1 - REVIEW

REVIEWER	Dr. Xie Sha Shenzhen University, Faculty of Education
REVIEW RETURNED	07-Apr-2023

GENERAL COMMENTS	Thank you for inviting me to review this manuscript. This paper is well-written and organized.
--

REVIEWER	Lizbeth Bullough
REVIEW RETURNED	17-Apr-2023

GENERAL COMMENTS	This paper addresses an important subject which is under-researched. It provides a robust theoretical and conceptual framework which is consistently maintained throughout. The data is publicly accessible, so consequently there are no ethical issues involved. The empirical data is rigorously examined and supports the argument. The potential impact of this research is wide ranging and beyond the remit of the academic world.
---

REVIEWER	Dr. Peter Flom Peter Flom Consulting
REVIEW RETURNED	20-Apr-2023

GENERAL COMMENTS	I confine my remarks to statistical and methodological aspects of this paper. The authors are clearly quite sophisticated users of statistics, but I have one big suggestion that, I think, would greatly improve the paper: Avoid using a dichotomous DV when it isn't necessary. Marking
---

	HBPPE as over 7% of total expenditures is a bad idea. It lowers power, increases type I error, and treats (e.g.) 0% and 6.9% as identical, but 6.9% as different from 7.1%. The sensitivity analysis helps with some of this, but it would be much better to leave PPE as a percentage, varying from 0% to whatever the highest value is. The authors could then use ordinary least squares (OLS) "regular" regression, or, perhaps, beta regression, if the bounds on the dependent variable violate the assumptions of the OLS model. (The authors should continue to use a multilevel model approach.) (A multilevel model for beta regression, if that is what the authors decide on, appears to be available in R see e.g. https://stats.stackexchange.com/questions/248045/how-to-implement-a-mixed-model-using-betareg-function-in-r) Given the high quality of the rest of the methods, I think maybe the authors had their reasons, but the ones listed in the paper are not, in my view, sufficient. After using OLS or beta regression, they could *present* results for different specific levels of PPE, I have no problem with that, but dichotomizing the variable does not seem warranted.
--	--

VERSION 1 – AUTHOR RESPONSE

Reviewer: 1

Thank you for inviting me to review this manuscript. This paper is well-written and organized.

Response: We appreciate the reviewer's comments on the paper.

Reviewer: 2

This paper addresses an important subject which is under-researched. It provides a robust theoretical and conceptual framework which is consistently maintained throughout. The data is publicly accessible, so consequently there are no ethical issues involved. The empirical data is rigorously examined and supports the argument. The potential impact of this research is wide ranging and beyond the remit of the academic world.

Response: We appreciate the reviewer's comments on the value of our study.

Reviewer: 3

I confine my remarks to statistical and methodological aspects of this paper. The authors are clearly quite sophisticated users of statistics, but I have one big suggestion that, I think, would greatly improve the paper:

Avoid using a dichotomous DV when it isn't necessary. Marking HBPPE as over 7% of total expenditures is a bad idea. It lowers power, increases type I error, and treats (e.g.) 0% and 6.9% as identical, but 6.9% as different from 7.1%. The sensitivity analysis helps with some of this, but it would be much better to leave PPE as a percentage, varying from 0% to whatever the highest value is. The authors could then use ordinary least squares (OLS) "regular" regression, or, perhaps, beta regression, if the bounds on the dependent variable violate the assumptions of the OLS model. (The authors should continue to use a multilevel model approach.) (A multilevel model for beta regression, if that is what the authors decide on, appears to be available in R see e.g. <https://stats.stackexchange.com/questions/248045/how-to-implement-a-mixed-model-using-betareg-function-in-r>)

Given the high quality of the rest of the methods, I think maybe the authors had their reasons, but the ones listed in the paper are not, in my view, sufficient. After using OLS or beta regression, they could *present* results for different specific levels of PPE, I have no problem with that, but dichotomizing the variable does not seem warranted.

Response: We understand the concern of the reviewer about the limitations of setting up thresholds for identifying HBPPE. The concern is fair and could apply to many measures using cut-offs, such as catastrophic health spending, extreme poverty, stunting, etc. However, using continuous variable won't serve a major purpose of this study. Our primary aim is to inform policy makers about who suffers from heavy financial burden for paying for PPE and needs financial help from governments. The idea of the cut-off is drawn from the concept of "catastrophic health spending," which is measured by the 10% or more of household total spending on health (World Bank's definition). The 10% cut-off for defining catastrophic health spending has been selected as an important policy measure for tracking the progress towards universal health coverage by the SDGs and many countries. This study advocates adopting a similar method so as to better understand household financial burdens for paying for PPE and hold governments accountable for their promise of "quality PPE for all girls and boys." (SDGs 4.2). If 10% of household spending on health has been widely accepted as "catastrophic" to the household, the same definition could be applied to PPE payments. Governments need to invest more in PPE to help poor households send their children to kindergartens.

Nevertheless, we agree with the reviewer about the shortcomings of using cut-offs and see the value of conducting analysis using continuous outcome variables. We therefore have added the analysis using proportion of household consumption on PPE as an outcome variable, as a part of the sensitivity tests. We have made revisions in the manuscript accordingly and added one chapter in the Appendix, including the related tables and figures (as presented below).

In the "Methods" section of the manuscript

"To further test the sensitivity of the findings, we used proportion of household consumption on PPE as an outcome variable and conducted descriptive and regression analysis, as shown in Chapter 5 of the Appendix."

In the "Results" section of the Manuscript

"Results from the sensitivity analysis using proportion of household spending on PPE as an outcome variable are presented in the Appendix (Figures A12-A14, Tables A8-A10). We used both a multilevel linear regression and a beta regression for estimation. As Tables A9 and A10 show, the results are consistent with the main regression results of using dichotomous outcome variable."

In the Appendix

Chapter 5. Sensitivity tests using proportion of households spending on PPE as the outcome variable and using continuous household net wealth as a predictor

We first calculated the provincial-level disparities in the % of household expenditure on PPE across the 29 provinces. On average, households in the Northeastern and middle areas spend higher % of household expenditure on PPE, with Shaanxi (15.4%) and Liaoning (14.8%) showing the highest levels. Hebei (6.4%) and Gansu (6.0%) had the lowest levels (Appendix Figure A12 and Table A8). The pattern across the provinces is similar to the % of households with HBPPE.

In terms of socioeconomic inequalities (Figure A13), the % of expenditure on PPE was higher among those households sending children to private kindergartens (10.5%) than to public kindergartens (8.1%). Statistically significant differences were not observed across urban and rural households, households with different levels of maternal education, or households in different wealth quintiles. The results are consistent with the ones in the text using dichotomous HBPPE.

Figure A14 shows results excluding out-of-school spending from the PPE measure. The difference between private and public kindergartens increases. Also the difference between the highest wealth quintile and the lowest one is statistically significant.

Tables A9 and A10 present the regression results using % of household total expenditure on PPE as the outcome variable. We used both a multilevel linear regression and a beta regression for estimation. The results are consistent with the main regression results in the text.

Figure A12. Level of % of household expenditure on PPE across 29 provinces

VERSION 2 – REVIEW

REVIEWER	Dr. Peter Flom Peter Flom Consulting
REVIEW RETURNED	16-Jun-2023

GENERAL COMMENTS	The authors have addressed my concerns and I now recommend publication.
---

VERSION 2 – AUTHOR RESPONSE

N/A